# Learning step sizes for unfolded sparse coding

**Pierre Ablin**[*]**, Thomas Moreau**[*]**, Mathurin Massias, Alexandre Gramfort**
Inria - CEA
Université Paris-Saclay
{pierre.ablin,thomas.moreau,mathurin.massias,alexandre.gramfort}@inria.fr

## Abstract

Sparse coding is typically solved by iterative optimization techniques, such as the Iterative Shrinkage-Thresholding Algorithm (ISTA). Unfolding and learning weights of ISTA using neural networks is a practical way to accelerate estimation. In this paper, we study the selection of adapted step sizes for ISTA. We show that a simple step size strategy can improve the convergence rate of ISTA by leveraging the sparsity of the iterates. However, it is impractical in most large-scale applications. Therefore, we propose a network architecture where only the step sizes of ISTA are learned. We demonstrate that for a large class of unfolded algorithms, if the algorithm converges to the solution of the Lasso, its last layers correspond to ISTA with learned step sizes. Experiments show that our method is competitive with state-of-the-art networks when the solutions are sparse enough.

## 1 Introduction

The resolution of convex optimization problems by iterative algorithms has become a key part of machine learning and signal processing pipelines, in particular with the Generalized Linear Models for classification [Nelder and Wedderburn, 1972]. Amongst these problems, special attention has been devoted to the Lasso [Tibshirani, 1996], due to the attractive sparsity properties of its solution (see Hastie et al. 2015 for an extensive review). For a given input $x \in \mathbb{R}^n$ , a dictionary $D \in \mathbb{R}^{n \times m}$ and a regularization parameter $\lambda > 0$ , the Lasso problem is

$$z^*(x) \in \underset{z \in \mathbb{R}^m}{\arg\min}\, F_x(z) \quad \text{with} \quad F_x(z) \triangleq \frac{1}{2}\|x - Dz\|^2 + \lambda\|z\|_1 \ . \tag{1}$$

A variety of algorithms exist to solve Problem (1), *e.g.* proximal coordinate descent [Tseng, 2001, Friedman et al., 2007], Least Angle Regression [Efron et al., 2004] or proximal splitting methods [Combettes and Bauschke, 2011]. The focus of this paper is on the Iterative Shrinkage-Thresholding Algorithm (ISTA, Daubechies et al. 2004), which is a proximal-gradient method applied to Problem (1). ISTA starts from $z^{(0)} = 0$ and iterates

$$z^{(t+1)} = \text{ST}\left(z^{(t)} - \frac{1}{L}D^\top(Dz^{(t)} - x), \frac{\lambda}{L}\right) \ , \tag{2}$$

where ST is the soft-thresholding operator defined as $\text{ST}(x, u) \triangleq \text{sign}(x)\max(|x| - u, 0)$ , and $L$ is the greatest eigenvalue of $D^\top D$ . In the general case, ISTA converges at rate $1/t$ , which can be improved to the *optimal* rate $1/t^2$ [Nesterov, 1983]. However, this optimality stands in the worst possible case, and linear rates are achievable in practice [Liang et al., 2014].

A popular line of research to improve the speed of Lasso solvers is to try to identify the support of $z^*$ , in order to diminish the size of the optimization problem [El Ghaoui et al., 2012, Ndiaye et al., 2017, Johnson and Guestrin, 2015, Massias et al., 2018].

---

[*]Equal contribution

Once the support is identified, larger steps can also be taken, leading to improved rates for first order algorithms [Liang et al., 2014, Poon et al., 2018, Sun et al., 2019].

However, these techniques only consider the case where a single Lasso problem is solved. When one wants to solve the Lasso for many samples $\{x^i\}_{i=1}^N$ – *e.g.* in dictionary learning [Olshausen and Field, 1997] – it is proposed by Gregor and Le Cun [2010] to *learn* a $T$-layers neural network of parameters $\Theta$, $\Phi_\Theta : \mathbb{R}^n \to \mathbb{R}^m$ such that $\Phi_\Theta(x) \simeq z^*(x)$. This Learned-ISTA (LISTA) algorithm yields better solution estimates than ISTA on new samples for the same number of iterations/layers. This idea has led to a profusion of literature (summarized in Table A.1 in appendix), and is a popular approach to solve inverse problems. Recently, it has been hinted by Zhang and Ghanem [2018], Ito et al. [2018], Liu et al. [2019] that only a few well-chosen parameters can be learned while retaining the performances of LISTA.

In this article, we study strategies for LISTA where only step sizes are learned. In Section 3, we propose Oracle-ISTA, an analytic strategy to obtain larger step sizes in ISTA. We show that the proposed algorithm's convergence rate can be much better than that of ISTA. However, it requires computing a large number of Lipschitz constants which is a burden in high dimension. This motivates the introduction of Step-LISTA (SLISTA) networks in Section 4, where only a step size parameter is learned per layer. As a theoretical justification, we show in Theorem 4.4 that the last layers of *any* deep LISTA network converging on the Lasso *must* correspond to ISTA iterations with learned step sizes. We validate the soundness of this approach with numerical experiments in Section 5.

## 2 Notation and Framework

**Notation**    The $\ell_2$ norm on $\mathbb{R}^n$ is $\|\cdot\|$. For $p \in [1, \infty]$, $\|\cdot\|_p$ is the $\ell_p$ norm. The Frobenius matrix norm is $\|M\|_F$. The identity matrix of size $m$ is $\mathrm{Id}_m$. ST is the soft-thresholding operator. Iterations are denoted $z^{(t)}$. $\lambda > 0$ is the regularization parameter. The Lasso cost function is $F_x$. $\psi_\alpha(z, x)$ is one iteration of ISTA with step $\alpha$: $\psi_\alpha(z, x) = \mathrm{ST}(z - \alpha D^\top(Dz - x), \alpha\lambda)$. $\phi_\theta(z, x)$ is one iteration of a LISTA layer with parameters $\theta = (W, \alpha, \beta)$: $\phi_\theta(z, x) = \mathrm{ST}(z - \alpha W^\top(Dz - x), \beta\lambda)$.

The set of integers between 1 and $m$ is $[\![1, m]\!]$. Given $z \in \mathbb{R}^m$, the support is $\mathrm{supp}(z) = \{j \in [\![1, m]\!] : z_j \neq 0\} \subset [\![1, m]\!]$. For $S \subset [\![0, m]\!]$, $D_S \in \mathbb{R}^{n \times m}$ is the matrix containing the columns of $D$ indexed by $S$. We denote $L_S$, the greatest eigenvalue of $D_S^\top D_S$. The equicorrelation set is $E = \{j \in [\![1, m]\!] : |D_j^\top(Dz^* - x)| = \lambda\}$. The equiregularization set is $\mathcal{B}_\infty = \{x \in \mathbb{R}^n : \|D^\top x\|_\infty = 1\}$. Neural networks parameters are between brackets, *e.g.* $\Theta = \{\alpha^{(t)}, \beta^{(t)}\}_{t=0}^{T-1}$. The sign function is $\mathrm{sign}(x) = 1$ if $x > 0$, $-1$ if $x < 0$ and 0 is $x = 0$.

**Framework**    This paragraph recalls some properties of the Lasso. Lemma 2.1 gives the first-order optimality conditions for the Lasso.

**Lemma 2.1** (Optimality for the Lasso). *The Karush-Kuhn-Tucker (KKT) conditions read*

$$z^* \in \arg\min F_x \Leftrightarrow \forall j \in [\![1, m]\!], D_j^\top(x - Dz^*) \in \lambda\partial|z_j^*| = \begin{cases} \{\lambda\,\mathrm{sign}\,z_j^*\}, & \text{if } z_j^* \neq 0, \\ [-\lambda, \lambda], & \text{if } z_j^* = 0. \end{cases} \tag{3}$$

Defining $\lambda_{\max} \triangleq \|D^\top x\|_\infty$, it holds $\arg\min F_x = \{0\} \Leftrightarrow \lambda \geq \lambda_{\max}$. For *some* results in Section 3, we will need the following assumption on the dictionary $D$:

**Assumption 2.2** (Uniqueness assumption). *$D$ is such that the solution of Problem* (1) *is unique for all $\lambda$ and $x$ i.e. $\arg\min F_x = \{z^*\}$.*

Assumption 2.2 may seem stringent since whenever $m > n$, $F_x$ is not strictly convex. However, it was shown in Tibshirani [2013, Lemma 4] – with earlier results from Rosset et al. 2004 – that if $D$ is sampled from a continuous distribution, Assumption 2.2 holds for $D$ with probability one.

**Definition 2.3** (Equicorrelation set). *The KKT conditions motivate the introduction of the equicorrelation set $E \triangleq \{j \in [\![1, m]\!] : |D_j^\top(Dz^* - x)| = \lambda\}$, since $j \notin E \implies z_j^* = 0$, i.e. $E$ contains the support of any solution $z^*$.*

*When Assumption 2.2 holds, we have $E = \mathrm{supp}(z^*)$ [Tibshirani, 2013, Lemma 16].*

We consider samples $x$ in the *equiregularization* set

$$\mathcal{B}_\infty \triangleq \{x \in \mathbb{R}^n : \|D^\top x\|_\infty = 1\} \ , \tag{4}$$

which is the set of $x$ such that $\lambda_{\max}(x) = 1$ . Therefore, when $\lambda \geq 1$ , the solution is $z^*(x) = 0$ for all $x \in \mathcal{B}_\infty$ , and when $\lambda < 1$ , $z^*(x) \neq 0$ for all $x \in \mathcal{B}_\infty$ . For this reason, we assume $0 < \lambda < 1$ in the following.

## 3   Better step sizes for ISTA

The Lasso objective is the sum of a $L$-smooth function, $\frac{1}{2}\|x - D \cdot\|^2$ , and a function with an explicit proximal operator, $\lambda\|\cdot\|_1$ . Proximal gradient descent for this problem, with the sequence of step sizes $(\alpha^{(t)})$ consists in iterating

$$z^{(t+1)} = \text{ST}\left(z^{(t)} - \alpha^{(t)}D^\top(Dz^{(t)} - x), \lambda\alpha^{(t)}\right) \ . \tag{5}$$

ISTA follows these iterations with a constant step size $\alpha^{(t)} = 1/L$ . In the following, denote $\psi_\alpha(z, x) \triangleq \text{ST}(z - \alpha D^\top(Dz^{(t)} - x), \alpha\lambda)$ . One iteration of ISTA can be cast as a majorization-minimization step [Beck and Teboulle, 2009]. Indeed, for all $z \in \mathbb{R}^m$ ,

$$F_x(z) = \tfrac{1}{2}\|x - Dz^{(t)}\|^2 + (z - z^{(t)})^\top D^\top(Dz^{(t)} - x) + \tfrac{1}{2}\|D(z - z^{(t)})\|^2 + \lambda\|z\|_1 \tag{6}$$

$$\leq \underbrace{\tfrac{1}{2}\|x - Dz^{(t)}\|^2 + (z - z^{(t)})^\top D^\top(Dz^{(t)} - x) + \tfrac{L}{2}\|z - z^{(t)}\|^2 + \lambda\|z\|_1}_{\triangleq Q_{x,L}(z, z^{(t)})} \ , \tag{7}$$

where we have used the inequality $(z - z^{(t)})^\top D^\top D(z - z^{(t)}) \leq L\|z - z^{(t)}\|^2$ . The minimizer of $Q_{x,L}(\cdot, z^{(t)})$ is $\psi_{1/L}(z^{(t)}, x)$, which is the next ISTA step.

**Oracle-ISTA: an accelerated ISTA with larger step sizes**   Since the iterates are sparse, this approach can be refined. For $S \subset [\![1, m]\!]$ , let us define the $S$-smoothness of $D$ as

$$L_S \triangleq \max_z z^\top D^\top Dz, \ \text{ s.t. } \|z\| = 1 \text{ and } \text{supp}(z) \subset S \ , \tag{8}$$

with the convention $L_\emptyset = L$ . Note that $L_S$ is the greatest eigenvalue of $D_S^\top D_S$ where $D_S \in \mathbb{R}^{n \times |S|}$ is the columns of $D$ indexed by $S$ . For all $S$ , $L_S \leq L$ , since $L$ is the solution of Equation (8) without support constraint. Assume $\text{supp}(z^{(t)}) \subset S$ . Combining Equations (6) and (8), we have

$$\forall z \text{ s.t. } \text{supp}(z) \subset S, \ F_x(z) \leq Q_{x,L_S}(z, z^{(t)}) \ . \tag{9}$$

The minimizer of the r.h.s is $z = \psi_{1/L_S}(z^{(t)}, x)$ . Furthermore, the r.h.s. is a tighter upper bound than the one given in Equation (7) (see illustration in Figure 1). Therefore, using $z^{(t+1)} = \psi_{1/L_S}(z^{(t)}, x)$ minimizes a tighter upper bound, provided that the following condition holds

$$\text{supp}(z^{(t+1)}) \subset S \ . \tag{$\star$}$$

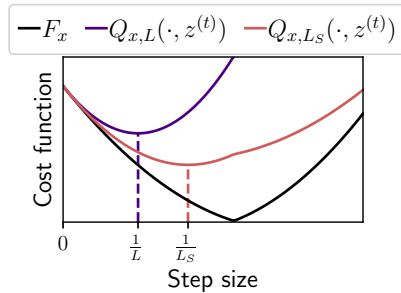

Figure 1: Majorization illustration. If $z^{(t)}$ has support $S$ , $Q_{x,L_S}(\cdot, z^{(t)})$ is a tighter upper bound of $F_x$ than $Q_{x,L}(\cdot, z^{(t)})$ on the set of points of support $S$ .

Oracle-ISTA (OISTA) is an accelerated version of ISTA which leverages the sparsity of the iterates in order to use larger step sizes. The method is summarized in Algorithm 1. OISTA computes

---

**Algorithm 1:** Oracle-ISTA (OISTA) with larger step sizes

---

**Input:** Dictionary $D$ , target $x$ , number of iterations $T$
$z^{(0)} = 0$
**for** $t = 0, \ldots, T-1$ **do**
    Compute $S = \mathrm{supp}(z^{(t)})$ and $L_S$ using an oracle ;
    Set $y^{(t+1)} = \psi_{1/L_S}(z^{(t)}, x)$ ;
    **if** *Condition* $\star$: $\mathrm{supp}(y^{(t+1)}) \subset S$ **then** Set $z^{(t+1)} = y^{(t+1)}$ ;
    **else** Set $z^{(t+1)} = \psi_{1/L}(z^{(t)}, x)$ ;
**Output:** Sparse code $z^{(T)}$

---

$y^{(t+1)} = \psi_{1/L_s}(z^{(t)}, x)$ , using the larger step size $1/L_s$ , and checks if it satisfies the support Condition $\star$. When the condition is satisfied, the step can be safely accepted. In particular Equation (9) yields $F_x(y^{(t+1)}) \leq F_x(z^{(t)})$ . Otherwise, the algorithm falls back to the regular ISTA iteration with the smaller step size. Hence, each iteration of the algorithm is guaranteed to decrease $F_x$ . The following proposition shows that OISTA converges in iterates, achieves finite support identification, and eventually reaches a safe regime where Condition $\star$ is always true.

**Proposition 3.1** (Convergence, finite-time support identification and safe regime). *When Assumption 2.2 holds, the sequence $(z^{(t)})$ generated by the algorithm converges to $z^* = \arg\min F_x$ .*

*Further, there exists an iteration $T^*$ such that for $t \geq T^*$ , $\mathrm{supp}(z^{(t)}) = \mathrm{supp}(z^*) \triangleq S^*$ and Condition $\star$ is always statisfied.*

*Sketch of proof (full proof in Subsection B.1).* Using Zangwill's global convergence theorem [Zangwill, 1969], we show that all accumulation points of $(z^{(t)})$ are solutions of Lasso. Since the solution is assumed unique, $(z^{(t)})$ converges to $z^*$ . Then, we show that the algorithm achieves finite-support identification with a technique inspired by Hale et al. [2008]. The algorithm gets arbitrary close to $z^*$ , eventually with the same support. We finally show that in a neighborhood of $z^*$ , the set of points of support $S^*$ is stable by $\psi_{1/L_S}(\cdot, x)$ . The algorithm eventually reaches this region, and then Condition $\star$ is true. $\qquad\square$

It follows that the algorithm enjoys the usual ISTA convergence results replacing $L$ with $L_{S^*}$ .

**Proposition 3.2** (Rates of convergence). *For $t > T^*$ , $F_x(z^{(t)}) - F_x(z^*) \leq L_{S^*} \frac{\|z^* - z^{(T^*)}\|^2}{2(t - T^*)}$ .*
*If additionally $\inf_{\|z\|=1} \|D_{S^*} z\|^2 = \mu^* > 0$ , then the convergence rate for $t \geq T^*$ is*
$F_x(z^{(t)}) - F_x(z^*) \leq (1 - \frac{\mu^*}{L_{S^*}})^{t - T^*} (F_x(z^{(T^*)}) - F_x(z^*))$ .

*Sketch of proof (full proof in Subsection B.2).* After iteration $T^*$ , OISTA is equivalent to ISTA applied on $F_x(z)$ restricted to $z \in S^*$ . This function is $L_{S^*}$-smooth, and $\mu^*$-strongly convex if $\mu^* > 0$ . Therefore, the classical ISTA rates apply with improved condition number. $\qquad\square$

These two rates are tighter than the usual ISTA rates – in the convex case $L \frac{\|z^*\|^2}{2t}$ and in the $\mu$-strongly convex case $(1 - \frac{\mu^*}{L})^t (F_x(0) - F_x(z^*))$ [Beck and Teboulle, 2009]. Finally, the same way ISTA converges in one iteration when $D$ is orthogonal ($D^\top D = \mathrm{Id}_m$), OISTA converges in one iteration if $S^*$ is identified and $D_{S^*}$ is orthogonal.

**Proposition 3.3.** *Assume $D_{S^*}^\top D_{S^*} = L_{S^*} \mathrm{Id}_{|S^*|}$ . Then, $z^{(T^*+1)} = z^*$ .*

*Proof.* For $z$ s.t. $\mathrm{supp}(z) = S^*$ , $F_x(z) = Q_{x, L_S}(z, z^{(T^*)})$ . Hence, the OISTA step minimizes $F_x$ . $\qquad\square$

**Quantification of the rates improvement in a Gaussian setting** The following proposition gives an asymptotic value for $\frac{L_S}{L}$ in a simple setting.

**Proposition 3.4.** *Assume that the entries of $D \in \mathbb{R}^{n \times m}$ are i.i.d centered Gaussian variables with variance $1$. Assume that $S$ consists of $k$ integers chosen uniformly at random in $[\![1, m]\!]$. Assume that $k, m, n \to +\infty$ with linear ratios $m/n \to \gamma, \ k/m \to \zeta$. Then*

$$\frac{L_S}{L} \to \left( \frac{1 + \sqrt{\zeta\gamma}}{1 + \sqrt{\gamma}} \right)^2 . \tag{10}$$

This is a direct application of the Marchenko-Pastur law [Marchenko and Pastur, 1967]. The law is illustrated on a toy dataset in Figure D.1. In Proposition 3.4, $\gamma$ is the ratio between the number of atoms and number of dimensions, and the average size of $S$ is described by $\zeta \leq 1$. In an overcomplete setting where we have $\gamma \gg 1$, this yield an approximation of Equation (10) with $L_S \simeq \zeta L$. Therefore, if $z^*$ is very sparse ($\zeta \ll 1$), the convergence rates of Proposition 3.2 are much better than those of ISTA.

**Backtracking Line Search**   A related strategy for finding good step sizes is the use of backtracking line search (see for instance Nesterov 2013). The core idea here is to compute iterate candidates for various step-sizes and choose the one that gives the best cost decrease. This strategy is adaptive to the actual state of the iterative procedure. However, it requires computing a new step size at each iteration. At each iteration, BT considers step-sizes of the form $(\alpha_0 \beta^k)_{k \geq 0}$, where $\alpha_0$ is an initial guess and $\beta < 1$ is a shrinking factor. In practice, the hyperparameters $\alpha_0$ and $\beta$ are critical and hard to tune. The need to search for a new step-size at each iteration is the main difference with OISTA which provides a fixed rule (maybe intractable) to set the step size.

**Example**   Figure 2 compares the OISTA, ISTA, FISTA, and backtracking ISTA on a toy problem. We display two backtracking strategies, with different hyperparameters. We also compare this to a greedy best step-size approach, where step-sizes are chosen as $\alpha^{(t+1)} = \arg\min F_x(\psi_\alpha(z^{(t)}, x))$. The improved rate of convergence of OISTA over ISTA and FISTA is illustrated: one can indeed take greater steps to increase the convergence speed. Further comparisons are displayed in Figure D.2 for different regularization parameters $\lambda$. While this demonstrates a faster rate of convergence, OISTA requires computing several Lipschitz constants $L_S$, which is cumbersome in high dimension. This motivates the next section, where we propose to *learn* those steps.

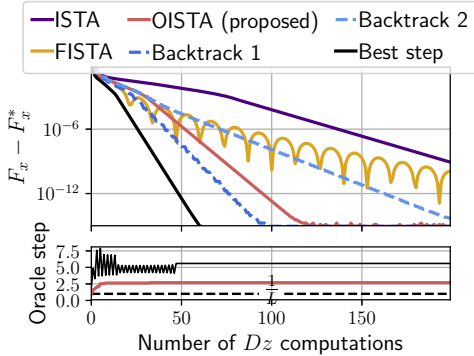

Figure 2: Convergence curves of OISTA, ISTA, FISTA, backtracking ISTA and a greedy best step-size strategy on a toy problem with $n = 10$, $m = 50$, $\lambda = 0.5$. The bottom figure displays the (normalized) steps taken by OISTA and the best steps at each iteration. Full experimental setup described in Appendix D.

## 4   Learning unfolded algorithms

**Network architectures**   At each step, ISTA performs a linear operation to compute an update in the direction of the gradient $D^\top (Dz^{(t)} - x)$ and then an element-wise non linearity with the soft-thresholding operator $\mathrm{ST}$. The whole algorithm can be summarized as a recurrent neural network (RNN), presented in Figure 3a. Gregor and Le Cun [2010] introduced Learned-ISTA (LISTA), a neural network constructed by unfolding this RNN $T$ times and learning the weights associated to each layer. The unfolded network, presented in Figure 3b, iterates $z^{(t+1)} = \mathrm{ST}(W_x^{(t)} x + W_z^{(t)} z^{(t)}, \lambda\beta^{(t)})$. It outputs exactly the same vector as $T$ iterations of ISTA when $W_x^{(t)} = \frac{D^\top}{L}$, $W_z^{(t)} = \mathrm{Id}_m - \frac{D^\top D}{L}$ and $\beta^{(t)} = \frac{1}{L}$. Empirically, this network is able to output a better estimate of the sparse code solution with fewer operations.

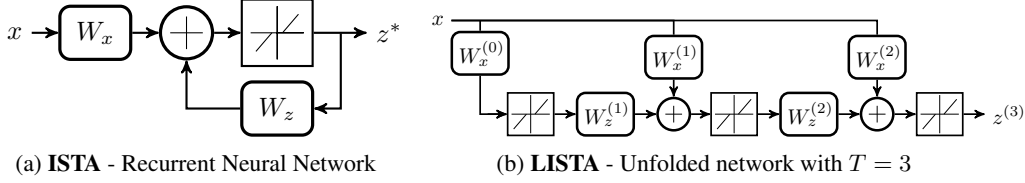

(a) **ISTA** - Recurrent Neural Network      (b) **LISTA** - Unfolded network with $T = 3$

Figure 3: Network architecture for ISTA (*left*) and LISTA (*right*).

Due to the expression of the gradient, Chen et al. [2018] proposed to consider only a subclass of the previous networks, where the weights $W_x$ and $W_z$ are coupled via $W_z = \mathrm{Id}_m - W_x^\top D$. This is the architecture we consider in the following. A layer of LISTA is a function $\phi_\theta : \mathbb{R}^m \times \mathbb{R}^n \to \mathbb{R}^m$ parametrized by $\theta = (W, \alpha, \beta) \in \mathbb{R}^{n \times m} \times \mathbb{R}_*^+ \times \mathbb{R}_*^+$ such that

$$\phi_\theta(z, x) = \mathrm{ST}(z - \alpha W^\top(Dz - x), \beta\lambda) \ . \tag{11}$$

Given a set of $T$ layer parameters $\Theta^{(T)} = \{\theta^{(t)}\}_{t=0}^{T-1}$, the LISTA network $\Phi_{\Theta^{(T)}} : \mathbb{R}^n \to \mathbb{R}^m$ is $\Phi_{\Theta^{(T)}}(x) = z^{(T)}(x)$ where $z^{(t)}(x)$ is defined by recursion

$$z^{(0)}(x) = 0, \quad \text{and} \quad z^{(t+1)}(x) = \phi_{\theta^{(t)}}(z^{(t)}(x), x) \quad \text{for } t \in [\![0, T-1]\!] \ . \tag{12}$$

Taking $W = D$, $\alpha = \beta = \frac{1}{L}$ yields the same outputs as $T$ iterations of ISTA.

To alleviate the need to learn the large matrices $W^{(t)}$, Liu et al. [2019] proposed to use a shared analytic matrix $W_{\mathrm{ALISTA}}$ for all layers. The matrix is computed in a preprocessing stage by

$$W_{\mathrm{ALISTA}} = \arg\min_W \|W^\top D\|_F^2 \quad s.t. \quad \mathrm{diag}(W^\top D) = \mathbf{1}_m \ . \tag{13}$$

Then, only the parameters $(\alpha^{(t)}, \beta^{(t)})$ are learned. This effectively reduces the number of parameters from $(nm + 2) \times T$ to $2 \times T$. However, we will see that ALISTA fails in our setup.

**Step-LISTA** With regards to the study on step sizes for ISTA in Section 3, we propose to *learn* approximation of ISTA step sizes for the input distribution using the LISTA framework. The resulting network, dubbed Step-LISTA (SLISTA), has $T$ parameters $\Theta_{\mathrm{SLISTA}} = \{\alpha^{(t)}\}_{t=0}^{T-1}$, and follows the iterations:

$$z^{(t+1)}(x) = \mathrm{ST}(z^{(t)}(x) - \alpha^{(t)} D^\top(Dz^{(t)}(x) - x), \alpha^{(t)}\lambda) \ . \tag{14}$$

This is equivalent to a coupling in the LISTA parameters: a LISTA layer $\theta = (W, \alpha, \beta)$ corresponds to a SLISTA layer if and only if $\frac{\alpha}{\beta}W = D$. This network aims at learning good step sizes, like the ones used in OISTA, without the computational burden of computing Lipschitz constants. The number of parameters compared to the classical LISTA architecture $\Theta_{\mathrm{LISTA}}$ is greatly diminished, making the network easier to train. Learning curves are shown in Figure D.3 in appendix.

Figure 4 displays the learned steps of a SLISTA network on a toy example. The network learns larger step-sizes as the sparsity (and as a result, $1/L_S$'s) increase. It is interesting to note that the learned step sizes tends to be larger than $1/L_S$ but smaller than $2/L_S$. As step sizes in $]0, 2/L_S[$ guarantee descent of the cost function, SLISTA learns step sizes that are adapted to solve the optimization problem. Still, steps larger than $2/L_S$ may be suitable depending on the geometry of the problem. For instance, in Figure 2, the greedy best-steps, that lead to the greatest decrease of the cost function, are taken larger than $2/L_S$.

**Training the network** We consider the framework where the network learns to solve the Lasso on $\mathcal{B}_\infty$ in an *unsupervised* way. Given a distribution $p$ on $\mathcal{B}_\infty$, the network is trained by solving

$$\tilde{\Theta}^{(T)} \in \arg\min_{\Theta^{(T)}} \mathcal{L}(\Theta^{(T)}) \triangleq \mathbb{E}_{x \sim p}[F_x(\Phi_{\Theta^{(T)}}(x))] \ . \tag{15}$$

Most of the literature on learned optimization train the network with a different *supervised* objective [Gregor and Le Cun, 2010, Xin et al., 2016, Chen et al., 2018, Liu et al., 2019]. Given a set of pairs $(x^i, z^i)$, the supervised approach tries to learn the parameters of the network such that $\Phi_\Theta(x^i) \simeq z^i$ *e.g.* by minimizing $\|\Phi_\Theta(x^i) - z^i\|^2$. This training procedure differs critically from ours. For instance, ISTA does not converge for the supervised problem in general while it does for the unsupervised one. As Proposition 4.1 shows, the unsupervised approach allows to *learn to minimize* the Lasso cost function $F_x$.

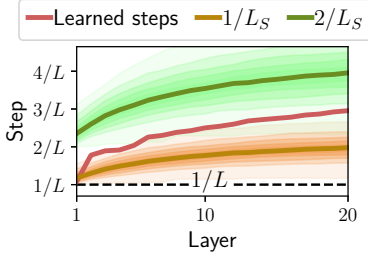

Figure 4: Steps learned with a 20 layers SLISTA network on a $10 \times 20$ problem. For each layer $t$ and each training sample $x$, we compute the support $S(x, t)$ of $z^{(t)}(x)$. The brown (resp. green) curves display the quantiles of the distribution of $1/L_{S(x,t)}$ (resp. $2/L_{S(x,t)}$) for each layer $t$. Learned steps are mostly in $]0, 2/L_S[$, which guarantees the decrease of the surrogate cost function. Full experimental setup described in Appendix D.

**Proposition 4.1** (Pointwise convergence). *Let $\tilde{\Theta}^{(T)}$ found by solving Problem* (15). *For $x \in \mathcal{B}_\infty$ such that $p(x) > 0$, $F_x(\Phi_{\tilde{\Theta}^{(T)}}(x)) \xrightarrow[T \to +\infty]{} F_x^*$ almost everywhere.*

*Sketch of proof (full proof in Subsection C.1).* Let $\Theta_{\text{ISTA}}^{(T)}$ the parameters corresponding to ISTA. For all $T$, we have $\mathbb{E}_{x \sim p}[F_x^*] \leq \mathbb{E}_{x \sim p}[F_x(\Phi_{\tilde{\Theta}^{(T)}}(x))] \leq \mathbb{E}_{x \sim p}[F_x(\Phi_{\Theta_{\text{ISTA}}^{(T)}}(x))]$. Because ISTA converges, the right hand term goes to $\mathbb{E}_{x \sim p}[F_x^*]$. Hence, $\mathbb{E}_{x \sim p}[F_x(\Phi_{\tilde{\Theta}^{(T)}}(x)) - F_x^*] \to 0$. This implies almost sure convergence of $F_x(\Phi_{\tilde{\Theta}^{(T)}}(x)) - F_x^*$ to 0 since it is non-negative. $\square$

**Asymptotical weight coupling theorem** In this paragraph, we show the main result of this paper: any LISTA network minimizing $F_x$ on $\mathcal{B}_\infty$ reduces to SLISTA in its deep layers (Theorem 4.4). It relies on the following Lemmas.

**Lemma 4.2** (Stability of solutions around $D_j$). *Let $D \in \mathbb{R}^{n \times m}$ be a dictionary with non-duplicated unit-normed columns. Let $c \triangleq \max_{l \neq j} |D_l^\top D_j| < 1$. Then for all $j \in [\![1, m]\!]$ and $\varepsilon \in \mathbb{R}^m$ such that $\|\varepsilon\| < \lambda(1 - c)$ and $D_j^\top \varepsilon = 0$, the vector $(1 - \lambda)e_j$ minimizes $F_x$ for $x = D_j + \varepsilon$.*

It can be proven by verifying the KKT conditions (3) for $(1 - \lambda)e_j$, detailed in Subsection C.2.

**Lemma 4.3** (Weight coupling). *Let $D \in \mathbb{R}^{n \times m}$ be a dictionary with non-duplicated unit-normed columns. Let $\theta = (W, \alpha, \beta)$ a set of parameters. Assume that all the couples $(z^*(x), x) \in \mathbb{R}^m \times \mathcal{B}_\infty$ such that $z^*(x) \in \arg\min F_x(z)$ verify $\phi_\theta(z^*(x), x) = z^*(x)$. Then, $\frac{\alpha}{\beta}W = D$.*

*Sketch of proof (full proof in Subsection C.3).* For $j \in [\![1, m]\!]$, consider $x = D_j + \varepsilon$, with $\varepsilon^\top D_j = 0$. For $\|\varepsilon\|$ small enough, $x \in \mathcal{B}_\infty$ and $\varepsilon$ verifies the hypothesis of Lemma 4.2, therefore $z^* = (1 - \lambda)e_j \in \arg\min F_x$. Writing $\phi_\theta(z^*, x) = z^*$ for the $j$-th coordinate yields $\alpha W_j^\top(\lambda D_j + \varepsilon) = \lambda\beta$. We can then verify that $(\alpha W_j^\top - \beta D_j^\top)(\lambda D_j + \varepsilon) = 0$. This stands for any $\varepsilon$ orthogonal to $D_j$ and of norm small enough. Simple linear algebra shows that this implies $\alpha W_j - \beta D_j = 0$. $\square$

Lemma 4.3 states that the Lasso solutions are fixed points of a LISTA layer only if this layer corresponds to a step size for ISTA. The following theorem extends the lemma by continuity, and shows that the deep layers of any converging LISTA network must tend toward a SLISTA layer.

**Theorem 4.4.** *Let $D \in \mathbb{R}^{n \times m}$ be a dictionary with non-duplicated unit-normed columns. Let $\Theta^{(T)} = \{\theta^{(t)}\}_{t=0}^{T}$ be the parameters of a sequence of LISTA networks such that the transfer function of the layer $t$ is $z^{(t+1)} = \phi_{\theta^{(t)}}(z^{(t)}, x)$. Assume that*

*(i) the sequence of parameters converges i.e. $\theta^{(t)} \xrightarrow[t \to \infty]{} \theta^* = (W^*, \alpha^*, \beta^*)$,*

*(ii) the output of the network converges toward a solution $z^*(x)$ of the Lasso* (1) *uniformly over the equiregularization set $\mathcal{B}_\infty$, i.e. $\sup_{x \in \mathcal{B}_\infty} \|\Phi_{\Theta^{(T)}}(x) - z^*(x)\| \xrightarrow[T \to \infty]{} 0$.*

*Then $\frac{\alpha^*}{\beta^*}W^* = D$.*

*Sketch of proof (full proof in Subsection C.4).* Let $\varepsilon > 0$, and $x \in \mathcal{B}_\infty$. Using the triangular inequality, we have

$$\|\phi_{\theta^*}(z^*, x) - z^*\| \leq \|\phi_{\theta^*}(z^*, x) - \phi_{\theta^{(t)}}(z^{(t)}, x)\| + \|\phi_{\theta^{(t)}}(z^{(t)}, x) - z^*\| \tag{16}$$

Since the $z^{(t)}$ and $\theta^{(t)}$ converge, they are valued over a compact set $K$. The function $f : (z, x, \theta) \mapsto \phi_\theta(z, x)$ is continuous, piecewise-linear. It is therefore Lipschitz on $K$. Hence, we have $\|\phi_{\theta^*}(z^*, x) - \phi_{\theta^{(t)}}(z^{(t)}, x)\| \leq \varepsilon$ for $t$ large enough. Since $\phi_{\theta^{(t)}}(z^{(t)}, x) = z^{(t+1)}$ and $z^{(t)} \to z^*$, $\|\phi_{\theta^{(t)}}(z^{(t)}, x) - z^*\| \leq \varepsilon$ for $t$ large enough. Finally, $\phi_{\theta^*}(z^*, x) = z^*$. Lemma 4.3 allows to conclude. □

Theorem 4.4 means that the deep layers of any LISTA network that converges to solutions of the Lasso correspond to SLISTA iterations: $W^{(t)}$ aligns with $D$, and $\alpha^{(t)}, \beta^{(t)}$ get coupled. This is illustrated in Figure 5, where a 40-layers LISTA network is trained on a $10 \times 20$ problem with $\lambda = 0.1$. As predicted by the theorem, $\frac{\alpha^{(t)}}{\beta^{(t)}} W^{(t)} \to D$ : the last layers only learn a step size. This is consistent with the observation of Moreau and Bruna [2017] which shows that the deep layers of LISTA stay close to ISTA. Further, Theorem 4.4 also shows that it is hopeless to optimize the unsupervised objective (15) with $W_{\text{ALISTA}}$ (13), since this matrix is not aligned with $D$.

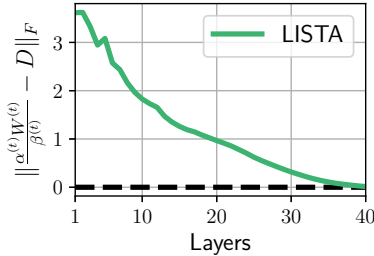

Figure 5: Illustration of Theorem 4.4: for deep layers of LISTA, we have $\alpha^{(t)} W^{(t)} / \beta^{(t)} \to D$, indicating that the network ultimately only learns a step size. Full experimental setup described in Appendix D.

# 5  Numerical Experiments

This section provides numerical arguments to compare SLISTA to LISTA and ISTA. All the experiments were run using Python [Python Software Foundation, 2017] and pytorch [Paszke et al., 2017]. The code to reproduce the figures is available online[2].

**Network comparisons**   We compare the proposed approach SLISTA to state-of-the-art learned methods LISTA [Chen et al., 2018] and ALISTA [Liu et al., 2019] on synthetic and semi-real cases.

In the synthetic case, a dictionary $D \in \mathbb{R}^{n \times m}$ of Gaussian i.i.d. entries is generated. Each column is then normalized to unit norm. A set of Gaussian i.i.d. samples $(\tilde{x}^i)_{i=1}^N \in \mathbb{R}^n$ is drawn. The input samples are obtained as $x^i = \tilde{x}^i / \|D^\top \tilde{x}^i\|_\infty \in \mathcal{B}_\infty$, so that for all $i$, $x^i \in \mathcal{B}_\infty$. We set $m = 256$ and $n = 64$.

For the semi-real case, we used the digits dataset from scikit-learn [Pedregosa et al., 2011] which consists of $8 \times 8$ images of handwritten digits from 0 to 9. We sample $m = 256$ samples at random from this dataset and normalize it do generate our dictionary $D$. Compared to the simulated Gaussian dictionary, this dictionary has a much richer correlation structure, which is known to imper the performances of learned algorithms [Moreau and Bruna, 2017]. The input distribution also consists from images from the digits dataset, normalized to lie in $\mathcal{B}_\infty$.

The networks are trained by minimizing the empirical loss $\mathcal{L}$ (15) on a training set of size $N_{\text{train}} = 10,000$ and we report the loss on a test set of size $N_{\text{test}} = 10,000$. Further details on training are in Appendix D.

Figure 6 shows the test curves for different levels of regularization $\lambda = 0.1$ and $0.8$. SLISTA performs best for high $\lambda$, even for challenging semi-real dictionary $D$. In a low regularization setting, LISTA performs best as SLISTA cannot learn much larger steps due to the low sparsity of the solution. In this unsupervised setting, ALISTA does not converge in accordance with Theorem 4.4.

# 6  Conclusion

We showed that using larger step sizes is an efficient strategy to accelerate ISTA for sparse solution of the Lasso. In order to make this approach practical, we proposed SLISTA, a neural network

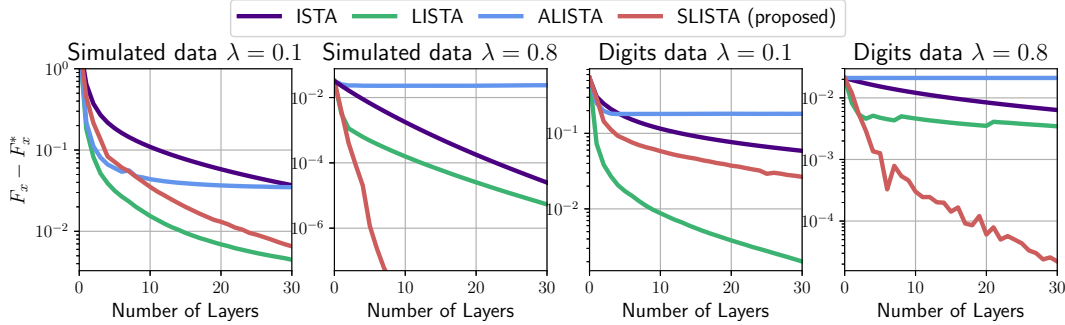

Figure 6: Test loss of ISTA, ALISTA, LISTA and SLISTA on simulated and semi-real data for different regularization parameters.

architecture which learns such step sizes. Theorem 4.4 shows that the deepest layers of any converging LISTA architecture must converge to a SLISTA layer. Numerical experiments show that SLISTA outperforms LISTA in a high sparsity setting. An major benefit of our approach is that it preserves the dictionary. We plan on leveraging this property to apply SLISTA in convolutional or wavelet cases, where the structure of the dictionary allows for fast multiplications.

## Acknowledgements

We would like to thank the anonymous reviewers for their insightful comments which have improved the quality of the paper. This project has received funding from the European Research Council (ERC) under the European Union's Horizon 2020 research and innovation program (Grant agreement No. 676943)

## Footnotes

[2] The code can be found at https://github.com/tomMoral/adopty

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
