[Supplementary Material]

# Learning step sizes for unfolded sparse coding

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

## A    Unfolded optimization algorithms literature summary

In Table A.1, we summarize the prolific literature on learned unfolded optimization procedures for sparse recovery. A particular focus is set on the chosen training loss training which is either supervised, with a regression of $z^i$ from the input $x^i$ for a given training set $(x^i, z^i)$, or unsupervised, where the objective is to minimize the Lasso cost function $F_x$ for each training point $x$.

Table A.1: Neural network for sparse coding

| Reference | Base Algo | Train Loss | Coupled weights | Remarks |
|---|---|---|---|---|
| Gregor and Le Cun (2010) | ISTA / CD | supervised | $\times$ | – |
| Sprechmann et al. (2012) | Block CD | unsupervised | $\times$ | Group $\ell_1$ |
| Sprechmann et al. (2013) | ADMM | supervised | N/A | – |
| Hershey et al. (2014) | NMF | supervised | $\times$ | NMF |
| Wang et al. (2015) | IHT | supervised | $\times$ | Hard-thresholding |
| Xin et al. (2016) | IHT | supervised | $\times/\checkmark$ | Hard-thresholding |
| Giryes et al. (2018) | PGD/IHT | supervised | N/A | Group $\ell_1$ |
| Yang et al. (2017) | ADMM | supervised | N/A | – |
| Adler et al. (2017) | ADMM | supervised | N/A | Wasserstein distance with $z^*$ |
| Borgerding et al. (2017) | AMP | supervised | $\times$ | – |
| Moreau and Bruna (2017) | ISTA | unsupervised | $\times$ | – |
| Chen et al. (2018) | ISTA | supervised | $\checkmark$ | Linear convergence rate |
| Ito et al. (2018) | ISTA | supervised | $\checkmark$ | MMSE shrinkage non-linearity |
| Zhang and Ghanem (2018) | PGD | supervised | $\checkmark$ | Sparsity of Wavelet coefficients |
| Liu et al. (2019) | ISTA | supervised | $\checkmark$ | Analytic weight $W_{\text{ALISTA}}$ |
| **Proposed** | ISTA | unsupervised | $\checkmark$ | – |

## B    Proofs of Section 3's results

### B.1    Proof of Proposition 3.1

We consider that the solution of the Lasso is unique, following the result of Tibshirani (2013)[Lemmas 4 and 16] when the entries of $D$ and $x$ come from a continuous distribution.

**Proposition 3.1** (Convergence, finite-time support identification and safe regime). *When Assumption 2.2 holds, the sequence $(z^{(t)})$ generated by the algorithm converges to $z^* = \arg\min F_x$ .*

*Further, there exists an iteration $T^*$ such that for $t \geq T^*$ , $\operatorname{supp}(z^{(t)}) = \operatorname{supp}(z^*) \triangleq S^*$ and Condition $\star$ is always statisfied.*

*Proof.* Let $z^{(t)}$ be the sequence of iterates produced by Algorithm 1. We have a *descent function*

$$F_x(z^{(t+1)}) - F_x(z^{(t)}) \leq -\frac{\gamma}{2}\|z^{(t+1)} - z^{(t)}\|^2 \leq -\frac{\min\|D_j\|}{2}\|z^{(t+1)} - z^{(t)}\|^2 \ , \qquad (17)$$

where $\gamma = L_S$ if Condition $\star$ is met, and $L$ otherwise. Additionally, the iterates are bounded because $F_x(z^{(t)})$ decreases at each iteration and $F_x$ is coercive. Hence we can apply Zangwill's Global Convergence Theorem (Zangwill, 1969). Any $z^*$ accumulation point of $(z^{(t)})_{t\in\mathbb{N}}$ is a minimizer of $F_x$ .

Since we only consider the case where the minimizer is unique, the bounded sequence $(z^{(t)})_{t\in\mathbb{N}}$ has a unique accumulation point, thus converges to $z^*$ .

The support identification is a simplification of a result of Hale et al. (2008), we include it here for completeness.

**Lemma B.1** (Approximation of the soft-thresholding). *Let $z \in \mathbb{R}, \nu > 0$ . For $\epsilon$ small enough, we have*

$$\mathrm{ST}(z + \epsilon, \nu) = \begin{cases} 0 \ , & \text{if } |z| < \nu \ , \\ \max(0, \epsilon)\, \mathrm{sign}(z) \ , & \text{if } |z| = \nu \ , \\ z + \epsilon - \nu \, \mathrm{sign}\, z \ , & \text{if } |z| > \nu \ . \end{cases} \tag{18}$$

Let $\rho > 0$ be such that Equation (18) holds for $\nu = \lambda/L$ , every $\epsilon < \rho$ , and every $z = z_j^* - \frac{1}{L}D_j^\top(Dz^* - x)$ .

Let $t \in \mathbb{N}$ such that $z^{(t)} = z^* + \epsilon$ , with $\|\epsilon\| \leq \rho$ . With $\epsilon' \triangleq (\mathrm{Id} - \frac{1}{L}D^\top D)\epsilon$ , we also have $\|\epsilon'\| \leq \rho$ . Let $j \in [\![1, m]\!]$ .

If $j \notin E$ , $|z_j^* - \frac{1}{L}D_j^\top(Dz^* - x)| = |\frac{1}{L}D_j^\top(Dz^* - x)| < \lambda/L$ hence $\mathrm{ST}(z_j^* - \frac{1}{L}D_j^\top(Dz^* - x) + \epsilon'_j, \lambda/L) = 0$ .

If $j \in E$ , $|z_j^* - \frac{1}{L}D_j^\top(Dz^* - x)| = |z_j^* + \frac{\lambda}{L}\mathrm{sign}\, z_j^*| > \lambda/L$ , and $\mathrm{sign}\,\mathrm{ST}(z_j^* - \frac{1}{L}D_j^\top(Dz^* - x) + \epsilon'_j, \lambda/L) = \mathrm{sign}\, z_j^*$ .

The same reasoning can be applied with $\rho'$ such that Equation (18) holds for $\nu = \lambda/L_{S^*}$ , every $\epsilon < \rho'$ , and every $z = z_j^* - \frac{1}{L_S^*}D_j^\top(Dz^* - x)$ . If we introduce $\eta > 0$ such that $\|\epsilon\| \leq \eta \implies \|(\mathrm{Id} - \frac{1}{L_{S^*}}D\top D)\epsilon\| \leq \rho'$ , in the ball of center $z^*$ and radius $\eta$ , the iteration with step size $L_{S^*}$ identifies the support.

Additionnally, since $\mathrm{Id} - \frac{1}{L_{S^*}}D_{S^*}^\top D_{S^*}$ is non expansive on vectors which support is $S^*$ , the iterations with the step $L_{S^*}$ never leave this ball once they have entered it.

Therefore, once the iterates enter $\mathcal{B}(z^*, \min(\eta, \rho))$ , Condition $\star$ is always satisfied.

$\square$

## B.2   Proof of Proposition 3.2

**Proposition 3.2** (Rates of convergence). *For $t > T^*$ , $F_x(z^{(t)}) - F_x(z^*) \leq L_{S^*}\frac{\|z^* - z^{(T^*)}\|^2}{2(t - T^*)}$ .*
*If additionally $\inf_{\|z\| = 1}\|D_{S^*}z\|^2 = \mu^* > 0$ , then the convergence rate for $t \geq T^*$ is*
$F_x(z^{(t)}) - F_x(z^*) \leq (1 - \frac{\mu^*}{L_{S^*}})^{t - T^*}(F_x(z^{(T^*)}) - F_x(z^*))$ .

*Proof.* For $t \geq T^*$ , the iterates support is $S^*$ and the objective function is $L_{S^*}$-smooth instead of $L$-smooth. It is also $\mu^*$ strongly convex if $\mu^* > 0$ . The obtained rates are a classical result of the proximal gradient descent method in these cases. $\square$

# C   Proof of Section 4's Lemmas

## C.1   Proof of Lemma 4.2

**Lemma 4.2** (Stability of solutions around $D_j$). *Let $D \in \mathbb{R}^{n \times m}$ be a dictionary with non-duplicated unit-normed columns. Let $c \triangleq \max_{l \neq j}|D_l^\top D_j| < 1$ . Then for all $j \in [\![1, m]\!]$ and $\varepsilon \in \mathbb{R}^m$ such that $\|\varepsilon\| < \lambda(1 - c)$ and $D_j^\top \varepsilon = 0$ , the vector $(1 - \lambda)e_j$ minimizes $F_x$ for $x = D_j + \varepsilon$ .*

*Proof.* Let $j \in [\![1, m]\!]$ and let $\varepsilon \in \mathbb{R}^m \cap D_j^\perp$ be a vector such that $\|\varepsilon\| < \lambda(1 - c)$ .
For notation simplicity, we denote $z^* = z^*(D_j - \varepsilon)$ .

$$D_j^\top(Dz^* - D_j - \varepsilon) = D_j^\top(-\lambda D_j - \varepsilon) = -\lambda = -\lambda\,\mathrm{sign}\, z_j^* , \tag{19}$$

since $1 - \lambda > 0$ . For the other coefficients $l \in [\![1, m]\!] \setminus \{j\}$ , we have

$$|D_l^\top (Dz^* - D_j - \varepsilon)| = |D_l^\top (-\lambda D_j - \varepsilon)| , \tag{20}$$

$$= |\lambda D_l^\top D_j + D_l^\top \varepsilon)| , \tag{21}$$

$$\leq \lambda |D_l^\top D_j| + |D_l^\top \varepsilon| , \tag{22}$$

$$\leq \lambda c + \|D_l\| \|\varepsilon\| , \tag{23}$$

$$\leq \lambda c + \|\varepsilon\| < \lambda , \tag{24}$$

$$\tag{25}$$

Therefore, $(1 - \lambda)e_j$ verifies the KKT conditions (3) and $z^*(D_j + \varepsilon) = (1 - \lambda)e_j$ . $\qquad\square$

## C.2   Proof of Lemma 4.3

**Lemma 4.3** (Weight coupling). *Let $D \in \mathbb{R}^{n \times m}$ be a dictionary with non-duplicated unit-normed columns. Let $\theta = (W, \alpha, \beta)$ a set of parameters. Assume that all the couples $(z^*(x), x) \in \mathbb{R}^m \times \mathcal{B}_\infty$ such that $z^*(x) \in \arg\min F_x(z)$ verify $\phi_\theta(z^*(x), x) = z^*(x)$. Then, $\frac{\alpha}{\beta}W = D$ .*

*Proof.* Let $x \in \mathcal{B}_\infty$ be an input vector and $z^*(x) \in \mathbb{R}^m$ be a solution for the Lasso at level $\lambda > 0$ . Let $j \in [\![1, m]\!]$ be such that $z_j^* > 0$ . The KKT conditions (3) gives

$$D_j^\top (Dz^*(x) - x) = -\lambda . \tag{26}$$

Suppose that $z^*(x)$ is a fixed point of the layer, then we have

$$\mathrm{ST}(z_j^*(x) - \alpha W_j^\top (Dz^*(x) - x), \lambda\beta) = z_j^*(x) > 0 . \tag{27}$$

By definition, $\mathrm{ST}(a, b) > 0$ implies that $a > b$ and $\mathrm{ST}(a, b) = a - b$ . Thus,

$$z_j^*(x) - \alpha W_j^\top (Dz^*(x) - x) - \lambda\beta = z_j^*(x) \tag{28}$$

$$\Leftrightarrow \quad \alpha W_j^\top (Dz^*(x) - x) + \lambda\beta = 0 \tag{29}$$

$$\Leftrightarrow \quad \alpha W_j^\top (Dz^*(x) - x) - \beta D_j^\top (Dz^*(x) - x) = 0 \qquad \text{by (26)} \tag{30}$$

$$\Leftrightarrow \quad (\alpha W_j - \beta D_j)^\top (Dz^*(x) - x) = 0 . \tag{31}$$

As the relation (31) must hold for all $x \in \mathcal{B}_\infty$ , it is true for all $D_j + \varepsilon$ for all $\varepsilon \in \mathcal{B}(0, \lambda(1-c)) \cap D_j^\perp$ . Indeed, in this case, $\|D^\top (D_j + \varepsilon)\|_\infty = 1$ . $D$ verifies the conditions of Lemma 4.2, and thus $z^* = (1 - \lambda)e_j$ , *i.e.*

$$(\alpha W_j - \beta D_j)^\top (D(1 - \lambda)e_j - (D_j + \varepsilon)) = 0 \tag{32}$$

$$(\alpha W_j - \beta D_j)^\top (-\lambda D_j - \varepsilon) = 0 \tag{33}$$

Taking $\varepsilon = 0$ yields $(\alpha W_j - \beta D_j)^\top D_j = 0$ , and therefore Eq. (33) becomes $(\alpha W_j - \beta D_j)^\top \varepsilon = 0$ for all $\varepsilon$ small enough and orthogonal to $D_j$ , which implies $\alpha W_j - \beta D_j = 0$ and concludes our proof. $\qquad\square$

## C.3   Proof of Theorem 4.4

**Theorem 4.4.** *Let $D \in \mathbb{R}^{n \times m}$ be a dictionary with non-duplicated unit-normed columns. Let $\Theta^{(T)} = \{\theta^{(t)}\}_{t=0}^T$ be the parameters of a sequence of LISTA networks such that the transfer function of the layer $t$ is $z^{(t+1)} = \phi_{\theta^{(t)}}(z^{(t)}, x)$ . Assume that*

*(i)   the sequence of parameters converges i.e. $\theta^{(t)} \xrightarrow[t\to\infty]{} \theta^* = (W^*, \alpha^*, \beta^*)$ ,*

*(ii)   the output of the network converges toward a solution $z^*(x)$ of the Lasso (1) uniformly over the equiregularization set $\mathcal{B}_\infty$ , i.e. $\sup_{x \in \mathcal{B}_\infty} \|\Phi_{\Theta^{(T)}}(x) - z^*(x)\| \xrightarrow[T\to\infty]{} 0$ .*

*Then $\frac{\alpha^*}{\beta^*}W^* = D$ .*

*Proof.* For simplicity of the notation, we will drop the $x$ variable whenever possible, *i.e.* $z^* = z^*(x)$ and $\phi_\theta(z) = \phi_\theta(z, x)$ . We denote $z^{(t)} = \Phi_{\Theta^{(t)}}(x)$ the output of the network with $t$ layers.

Let $\epsilon > 0$ . By hypothesis (i), there exists $T_0$ such that for all $t \geq T_0$ ,

$$\|W^{(t)} - W^*\| \leq \epsilon \quad |\alpha^{(t)} - \alpha^*| \leq \epsilon \quad |\beta^{(t)} - \beta^*| \leq \epsilon . \tag{34}$$

By hypothesis (ii), , there exists $T_1$ such that for all $t \geq T_1$ and all $x \in \mathcal{B}_\infty$ ,

$$\|z^{(t)} - z^*\| \leq \epsilon . \tag{35}$$

Let $x \in \mathcal{B}_\infty$ be an input vector and $t \geq \max(T_0, T_1)$ . Using (35), we have

$$\|z^{(t+1)} - z^{(t)}\| \quad \leq \quad \|z^{(t+1)} - z^*\| + \|z^{(t)} - z^*\| \leq 2\epsilon \tag{36}$$

By (i), there exist a compact set $\mathcal{K}_1 \subset \mathbb{R}^{n \times m} \times \mathbb{R}_*^+ \times \mathbb{R}_*^+$ *s.t.* $\theta^{(t)} \in \mathcal{K}_1$ for all $t \in \mathbb{N}$ and $\theta^* \in \mathcal{K}$ . The input $x$ is taken in a compact set $\mathcal{B}_\infty$ and as $z^* = \arg\min_z F_x(z)$ , we have $\lambda \|z\|_1 \leq F_x(z^*) \leq F_x(0) = \|x\|$ thus $z^*$ is also in a compact set $\mathcal{K}_2$ .

We consider the function $f(z, x, \theta) = \mathrm{ST}(z - \alpha W^\top(Dz - x), \beta)$ on the compact set $\mathcal{K}_2 \times \mathcal{B}_\infty \times \mathcal{K}_1$ .

This function is continuous and piece-wise linear on a compact set. It is thus $L$-Lipschitz and thus

$$\|\phi_{\theta^{(t)}}(z^{(t)}) - \phi_{\theta^{(t)}}(z^*)\| \quad \leq \quad L\|z^{(t)} - z^*\| \leq L\epsilon \tag{37}$$

$$\|\phi_{\theta^*}(z^*) - \phi_{\theta^{(t)}}(z^*)\| \quad \leq \quad L\|\theta^{(t)} - \theta^*\| \leq L\epsilon \tag{38}$$

Using these inequalities, we get

$$\|\phi_{\theta^*}(z^*, x) - z^*\| \quad \leq \quad \underbrace{\|\phi_{\theta^*}(z^*) - \phi_{\theta^{(t)}}(z^*)\|}_{<L\epsilon \text{ by (38)}} + \underbrace{\|\phi_{\theta^{(t)}}(z^*) - \phi_{\theta^{(t)}}(z^{(t)})\|}_{<L\epsilon \text{ by (37)}} \tag{39}$$

$$+ \underbrace{\|\phi_{\theta^{(t)}}(z^{(t)}) - z^{(t)}\|}_{<2\epsilon \text{ by (36)}} + \underbrace{\|z^{(t)} - z^*\|}_{<\epsilon \text{ by (35)}}$$

$$\leq \quad (2L + 3)\epsilon . \tag{40}$$

As this result holds for all $\epsilon > 0$ and all $x \in \mathcal{B}_\infty$ , we have $\phi_{\theta^*}(z^*) = z^*$ for all $x \in \mathcal{B}_\infty$ . We can apply the Lemma 4.3 to conclude this proof. $\qquad\square$

# D    Experimental setups and supplementary figures

**Dictionary generation**: Unless specified otherwise, to generate synthetic dictionaries, we first draw a random i.i.d. Gaussian matrix $\hat{D} \in \mathbb{R}^{n \times m}$. The dictionary is obtained by normalizing the columns: $D_{ij} = \frac{1}{\|\hat{D}_{i:}\|}\hat{D}_{ij}$.

**Samples generation**: The samples $x$ are generated as follows: Random i.i.d. Gaussian samples $\hat{x} \in \mathbb{R}^n$ are generated. We then normalize them: $x = \frac{1}{\|D^\top \hat{x}\|_\infty}\hat{x}$, so that $x \in \mathcal{B}_\infty$.

**Training the networks** Since the loss function and the network are continuous but non-differentiable, we use sub-gradient descent for training. The sub-gradient of the cost function with respect to the parameters of the network is computed by automatic differentiation. We use full-batch sub-gradient descent with a backtracking procedure to find a suitable learning rate. To verify that we do not overfit the training set, we always check that the test loss and train loss are comparable.

**Main text figures setup**

- Figure 2: We generate a random dictionary of size $10 \times 50$. We take $\lambda = 0.5$, and a random sample $x \in \mathcal{B}_\infty$. $F_x^*$ is computed by iterating ISTA for 10000 iterations.

- Figure 4: We generate a random dictionary of size $10 \times 20$. We take $\lambda = 0.2$. We generate a training set of $N = 1000$ samples $(x^i)_{i=1}^{1000} \in \mathcal{B}_\infty$. A 20 layers SLISTA network is trained by gradient descent on these data. We report the learned step sizes. For each layer $t$ of the network and each training sample $x$, we compute the support at the output of the $t$-th layer, $S(x, t) = \mathrm{supp}(z^{(t)}(x))$. For each $t$, we display the quantiles of the distribution of the $(1/L_{S(x^i, t)})_{i=1}^{1000}$.

- Figure 5: A random $10 \times 20$ dictionary is generated. We take 1000 training samples, and $\lambda = 0.05$. A 40 layers LISTA network is trained by gradient descent on those samples. We report the quantity $\|\alpha^{(t)}W^{(t)} - \beta^{(t)}D\|_F$ for each layer $t$.

 **Supplementary experiments**

Figure D.1: Illustration of Proposition 3.4. A toy Gaussian dictionary is generated with $n = 200$, $m = 600$ so that $\gamma = 3$. We compute its Lipschitz constant $L$. For $\zeta$ between 0 and 1, we extract $\lfloor \zeta m \rfloor$ columns at random and compute the corresponding Lipschitz constant $L_S$. The plot shows an almost perfect fit between the empirical law and the theoretical limit (10).

Figure D.2: Comparison between ISTA, FISTA and Oracle-ISTA for different levels of regularization on a Gaussian dictionnary, with $n = 100$ and $m = 200$. We report the average number of iterations taken to reach a point $z$ such that $F_x(z) < F_x^* + 10^{-13}$. The experiment is repeated 10 times, starting from random points in $\mathcal{B}_\infty$. OISTA is always faster than ISTA, and is faster than FISTA for high regularization.

Figure D.3: Learning curves of SLISTA and LISTA. Random Gaussian dictionaries with $n = 10$ and $m = 20$ are generated. We take $\lambda = 0.3$. Networks with 10 layers are fit on those dictionaries, and their test loss is reported for different number of training samples. The process is repeated 100 times; the curves shown display the median of the test-loss.