[Reviews · NeurIPS 2019]

Reviewer 1



The proposed method of step size learning for ISTA with fast convergence, as well as the extension to LISTA step size approximation is novel and interesting. The proposed methods are all theoretically grounded, with detailed analysis. The presentation is also clear. The paper can be further improved with more numerical experiments to demonstrate the effectiveness of the learned model.

Reviewer 2



Update: I feel the authors satisfactory addressed my comments: they now included comparisons with back-tracking, showing indeed that it is on par (or perhaps better) than their Oracle-ISTA. This is an honest clarification that I appreciate. While they did not write the entire proof of prop. 4.1, I can now see how this could be made correct and clearer, and I look forward to seeing the full detailed proof. I also appreciate their other smaller modifications (numerical experiments). ---------------------------------------------------------------------------------- While this is an interesting paper, easy to read and with new contributions, there are a few points that are somewhat troubling. My main concerns regard the following points: 1) The improvement of convergence speed by increasing the step-size due to the observation that not all atoms are used is natural. Yet, there exist other alternatives to do the same: one could simply employ ISTA with back-tracking. If there are larger step-sizes to be employed, such an alternative would find them too. Why then should someone use OISTA instead of simply back-tracking? To my understanding, this point deserves a discussion and numerical comparison when appropriate (say, for Figure 2). 2) The conclusion of the authors seems to be that all is needed for acceleration is to learn better step-sizes. However, by greatly limiting the number of parameters that are learned, one must be certainly incurring in some restrictions. For instance, the work in [2] shows that when a particular factorization of the dictionary Gram is possible, acceleration is obtained. Such an acceleration would not be possible in the proposed SLISTA. Could the authors comment on this? 3) It is interesting that the authors adopt an unsupervised approach for learning the step-sizes. But then again: why not simply set them through backtracking? This would not require any learning, it is conceptually simpler, and would likely be faster. 4) Proposition 4.1 is not clear to me. In the proof, the authors employ the convergence of ISTA. However, this refers to the convergence of the iterates, and has little to do with the convergence of the learning problem in Eq (15). More generally, the authors mention (line 182) that the learning converges, while the loss (15) is a non-convex function of the parameters (\alpha^t). Could the authors explain? Further comments: - The authors seem to refer (most times) to the overcomplete case where m>n. However, this is not mentioned anywhere, and it is not a minor detail. Note, for instance, that if n>>m, the iterates will not be sparse. Consider perhaps just adding a comment on this in the beginning. - In Eq (8) (and beyond), shouldn't $supp(z)\subseteq S$? - line 136: saying that $\gamma \gg 1$ is a bit of a stretch: in a typical redundant case, $\gamma = 2-5$. In addition, consider mentioning that when n>m, k might be equal to m and so Ls/L \approx 1. - In Figure 4, the authors show that the learned step-sizes are larger than the oracle ones obtained by computing Ls. This seems to imply that the learned approach provides even faster convergence than the Oracle ISTA (because \alpha(t)>1/Ls). I might be missing something, but how come this is possible without diverging? - In the experiments section (5), the authors study a semi-real case by creating a dictionary of (8x8) digit images. The input signals seem to be constructed as (normalized) Gaussian noise samples (line 244). This makes no sense to me - digit atoms will not be able to sparsely represent noise. In fact, the constructed dictionary will likely not even span R^{64}. A more sensible approach would have been to take as input other samples from the dataset not included in the dictionary. - line 188: "any compact ..." domain? - line 237: normalized to 'unit norm'? [1] Moreau, Thomas, and Joan Bruna. "Understanding trainable sparse coding via matrix factorization." ICLR

Reviewer 3



The paper utilizes the fact that when the support of sparse code is identified, larger step size can be used to speed up the convergence. If I understand it correctly, as alpha/beta*W = D, then alpha*W in (11) is replaced by beta*D, and the authors claim that only tuning beta (which is replaced to alpha in (14)) is enough. However, in the experiments, there're no results on the effects of different step sizes. Another issue is that in section 5, I cannot tell why SLISTA is better. And I am curious about why ALISTA is nearly not working at all. Also, the authors mention that this LISTA style model can be trained unsupervised or supervised, are there any empirical evaluation on that? In fact, the claim that ISTA does not converge for the supervised problem in general seems odd. Is there any support for this claim? Finally, in comparison to ALISTA, this paper is less general and comprehensively analyzed. For example, the convolutional LISTA and robust version are interesting to consider.

[Author Response · NeurIPS 2019]

We thank the reviewers for their careful reading and their constructive comments. We clarify the main points raised in the reviews:

**Rev#1** **1.1 More experiments**: With suggestion from Rev.#2, we propose to replace the randomly generated $x$ in Figure.6(c/d) by images from the digits dataset. The curves do not change significantly but this use-case is close to a real application. Also, such method could be used to accelerate the resolution of inverse problems (*e.g.* see Adler et al 2017). We will emphasize such application in our introduction.

**Rev#2** **2.1 Comparison with backtracking (BT)**: BT typically considers (e.g. Nesterov 2013) candidate steps of the form $\alpha_0 \beta^k$, $k \geq 0$, where $\alpha_0$ is an initial guess and $\beta < 1$ is a shrinking constant (Armijo line-search). It is indeed close to our work with two main differences: i) At test time, BT should be performed at each step of ISTA, making the cost of one iteration larger than ISTA/SLISTA. Also note that the hyperparameters $\alpha_0, \beta$ are critical and hard to tune properly. ii) BT finds greedily a suitable step-size at *a given iteration* for *a given sample*. The goal with SLISTA is to learn a sequence of step-sizes for *all iterations* jointly for the *input distribution*. We will clarify this relationship and core differences in the text and replace Fig.2 with Fig A.2, where we show the results of a perfect line-search algorithm (taking at each iteration a step-size exactly minimizing the loss function), and two backtracking with different parameters $(\alpha_0, \beta)$. One could also use BT to learn steps for the whole distribution, by picking step sizes in the set $(\alpha_0 \beta^k)_{k \geq 0}$. This corresponds to a discretized version of the SLISTA problem, which we feel is out of the scope of the current article.

**2.2 Results relative to [2]**: We stress that Theorem 4.4 states that LISTA only learns step-sizes *asymptotically*. Thus, LISTA can still leverage the dictionary structure in the first layers and improve compared to SLISTA, as seen in Fig.6(a/c). Our results are consistent with those of [2], as their section 2.3.1 shows a phase transition where the structure of $D$ leveraged by FacNet only improves the convergence *in the first layers*.

**2.3 Clarify proposition 4.1**: The proof will be rewritten as the part about uniform convergence of ISTA is indeed confusing. ISTA's rate of convergence writes: $F_x(z_T(x)) - F_x^* \leq \frac{L}{T}\|z^*(x)\|_2^2$. Further, $\|z^*\|_2 \leq \|z^*\|_1 \leq \frac{1}{\lambda}F_x^* \leq \frac{1}{\lambda}F_x(0) = \frac{1}{\lambda}\|x\|_2^2$. Since $\mathcal{B}_\infty$ is bounded, $\|z^*(x)\|_2$ is uniformly bounded. It yields an inequality of the form $F_x^* \leq F_x(z_T(x)) \leq F_x^* + K/T$ where $K$ is independent of $x$. Taking expectations shows as advertised: $\mathbb{E}_{x \sim p}[F_x(\Phi_{\Theta_{\text{ISTA}}^{(T)}}(x))] \xrightarrow[T \to \infty]{} \mathbb{E}_{x \sim p}[F_x^*]$. This proposition shows the global (*theoretical*) minimizer of the unsupervised problem solves the LASSO. In practice, as any neural network, the computed solution faces optimization (non-convexity) and generalization (empirical risk minimization) errors.

**2.4 Fig.4:** We modify Fig.4 by adding the $2/L_S$ line (see Fig A.1). Learned steps are mainly included in $]0, 2/L_S[$ which guarantees the cost function decrease, if the support inclusion condition is verified. However, steps above $2/L_S$ may lead to greater decrease of the loss function as seen in Fig A.2.

**2.5 Semi-real experiment**: see 1.1.

**Rev#3** **3.1 Effect of different step-sizes**: Fig.2 shows that larger step-sizes can lead to faster convergence. See 2.1.

**3.2 Is SLISTA better? ALISTA does not work** In the experiments, we see that SLISTA is better when the iterates are very sparse (high $\lambda$), leveraging the same properties as OISTA. It is outperformed by LISTA for small $\lambda$. As stated in the text (l.229/252), following thm.4.4., ALISTA cannot converge in this unsupervised setting.

**3.3 Unsupervised/Supervised**: ISTA converges to a solution of the Lasso. In the supervised setting, a unique solution exists independently from the Lasso solution. In most practical cases, the Lasso solution and the supervised solution are different. Even if they match, it is for a specific $\lambda$, unkown *a piori*. Hence ISTA does not converge for the supervised problem. This is typically highlighted in Figure.1(a) from the ALISTA article (Liu et al 2019) where the MSE of ISTA plateaus. We will clarify this in the text.

**3.4 Extension to convolutional cases/robustness:** This is surely an interesting extension, which cannot unfortunately be inserted in the paper for lack of space. Also, the theoretical results from ALISTA paper were previously published in NeurIPS without the convolution/robustness part (Chen et al 2018).

(A.1) Step-sizes learned with SLISTA

(A.2) Performance of F/ISTA, OISTA and ISTA with Backtracking / oracle line search.

(A.3) Performances of SLISTA on digits with $\lambda = 0.8$.

[Meta-Review · NeurIPS 2019]

Originally, the paper received three borderline scores 7/6/5, with acceptable confidences 2/4/3. During discussion, two of the reviewers found that the rebuttal convinced them and both raised their scores. The AC agreed to the reviewers that the idea is novel but the paper needs many imprvements. So the AC recommend acceptance.